# Proportional Allocation of Indivisible Resources under Ordinal and Uncertain Preferences

**Zihao Li**[1]        **Xiaohui Bei**[1]        **Zhenzhen Yan**[1]

[1]School of Physical and Mathematical Sciences, Nanyang Technological University, Singapore

## Abstract

We study a fair resource allocation problem with indivisible items. The agents' preferences over items are assumed to be ordinal and have uncertainties. We adopt stochastic dominance proportionality as our fairness notion and study a sequence of problems related to finding allocations that are fair with a high probability. We provide complexity analysis for each problem and efficient algorithms for some problems. Finally, we propose several heuristic algorithms to find an allocation that is fair with the highest probability. We thoroughly evaluate the performance of the algorithms on both synthetic and real datasets.

## 1 INTRODUCTION

The problem of fairly allocating a set of scarce resources among multiple agents has been a central research topic in multi-agent systems and AI. One of the most prominent fairness notions in fair division is *proportionality*, which states that each agent should receive enough resources which are at least $1/n$ of the utility when she receives all the resources, where $n$ is the total number of the agents. Proportionality is one of the first studied fairness concepts in the literature. The existence of a proportional allocation can always be guaranteed when the resources are divisible. However, with indivisible items, proportional allocations may fail to exist in some cases; a simple example is when there are only two agents and one item which both agents valued positively.

A classic notion of proportionality is defined under the setting that each agent has a *cardinal* and *deterministic* valuation function over the items. In contrast, this paper studies *ordinal* and *uncertain* preferences over items. Considering only ordinal preferences is a natural and appealing assumption due to several reasons. To name a few, first, it is often much easier to elicit ordinal preferences from the agents since it requires significantly less information. Second, some weaker fairness notions defined for ordinal preferences may permit more positive computational results for many problems. Finally, in certain applications, it's not available to get the cardinal preferences due to agents' limited knowledge on their valuations. In fact, the ordinal preferences have been widely studied in the resource allocation problems [Kuhn, 1955, Gardenfors, 1973, Demko and Hill, 1988, Bogomolnaia and Moulin, 2001, Burkard et al., 2009, David, 2013, Aziz et al., 2019b].

The assumption of uncertain preferences is also practical. In some settings the ordinal preferences may not be completely known due to a lack of information or a high cost to elicit a full preference list. Specifically, it may require a large number of pairwise comparisons among possible options to get the full preference order. It can also be difficult to even rank two close options without additional costly information [Drummond and Boutilier, 2014]. In addition, according to the study of uncertain preferences in [Aziz et al., 2019a], an agent's preference may be a composition of several preference lists from other agents she represents, or a combination of preferences from different criteria, where their weights are not determined.

In summary, the problem studied in this paper is to find a fair allocation for agents that have ordinal and uncertain preferences over items. In particular, the notion to define fairness under ordinal preferences is based on *stochastic dominance (SD)*, a standard way of comparing fractional allocations. We model the uncertain preferences following a similar way as in Aziz et al. [2019a] and Aziz et al. [2016]. Specifically, we use tied sets to represent the items that the agents are not sure of their preferences towards and assume equal probabilities over each linear order extension of the preference order in the tied sets. This model of uncertainty is common for ordinal preferences. It has also been widely used in the matching literature [Rastegari et al., 2013, 2014, Drummond and Boutilier, 2014]. To the best of our knowledge, this paper is the first one to consider fair allocations under such preference settings and provide algorithmic and

*Accepted for the 38th Conference on Uncertainty in Artificial Intelligence* (UAI 2022).

complexity analysis for it.

## 1.1 OUR RESULTS

In this work, we study how to find proportionally fair allocations when the agents have ordinal and uncertain preferences. Inspired by Aziz et al. [2017b] which studies Pareto optimal allocations under similar preference models, we study the computational complexity of the following four problems:

- FAIRPROB: What is the probability that a given allocation is fair under the specific fairness definition?

- EXISTSPOSSIBLYFAIR: Does there exist an allocation that is fair with non-zero probability?

- EXISTSCERTAINLYFAIR: Does there exist an allocation that is fair with probability one?

- HIGHESTPROB: How to find an allocation with the highest probability of being fair under the specific fairness definition?

We focus on two fairness notions introduced in Aziz et al. [2015]: weak SD proportionality and SD proportionality. Other proportional notions are also discussed in Aziz et al. [2015]. But they are shown to be equivalent to either of these two fairness notions. For each fairness notion, we present hardness results and efficient algorithms for the above four questions. Specifically, we show that for both fairness notions, FAIRPROB and EXISTSPOSSIBLYFAIR can be solved in polynomial time via dynamic programming and matching. EXISTSCERTAINLYFAIR and HIGHESTPROB are NP-hard. But we can provide polynomial-time algorithms for some special cases. For instance, if the number of agents is constant or the agents are all identical, we can efficiently find an allocation satisfying the corresponding condition. We summarize all the theoretical findings in Table 1.

Finally, noted the importance of HIGHESTPROB in many real-world scenarios, we propose several heuristic algorithms to find an allocation that is fair with a high probability. Experiments in both synthetic and real datasets are conducted to evaluate their performance and computation efficiency.

## 1.2 RELATED WORKS

In most existing literature, agents are assumed to have *cardinal* and *deterministic* valuations over items [Lipton et al., 2004, Brams et al., 2012, Bouveret and Lang, 2008, Bezáková and Dani, 2005].

Several works have considered ordinal preferences and weaker fairness notions [Kuhn, 1955, Gardenfors, 1973, Demko and Hill, 1988, Bogomolnaia and Moulin, 2001,

Burkard et al., 2009, David, 2013, Aziz et al., 2015, Segal-Halevi et al., 2020, Bouveret et al., 2010]. Aziz et al. [2015] is the most related one. They used the stochastic dominance relation between fractional allocations to define various generalized notions of proportionality and envy-freeness and investigated the computational complexity of finding a fair assignment. They assume preferences are completely known and consider possibly indifferent preferences over items. In contrast, we assume that agents have strictly different preferences for different items but their preferences may not be completely known. We name the set of items with unknown preferences a tie. We model the preferences for items in a tie using uniformly distributed linear order extensions. It is worthwhile to note that assuming indifference for items with unknown preference can be regarded as an alternative way to model the uncertain preference. But we show that the fair allocation from such a modeling perspective can only be mapped to the certainly and possibly fair allocation in weak SD proportionality and SD proportionality, respectively in our work, after some small modifications.

Another stream of related literature considers uncertainties in agents' knowledge about their valuations [Lumet et al., 2012, Aziz et al., 2016, 2017b,a, 2019a, Menon and Larson, 2018]. The most related ones are Aziz et al. [2019a] and Aziz et al. [2017b], in which they study a Pareto optimal assignment problem. In their problem, agents also express ordinal and uncertain preferences. We adopt the way how they model the uncertain preferences but study a proportionally fair allocation problem. Lumet et al. [2012] considered a different fair division uncertainty model in which each item could be in either good or bad state with certain probabilities. They developed algorithms to find ex-post fair allocations assuming independent states over items and correlated valuations over agents. Different from their paper, we assume the states of the items are correlated but the agent valuations are independent.

## 2 PRELIMINARIES

Consider an allocation problem instance with a set of $n$ agents $N = \{1, \ldots, n\}$ and a set of $m$ indivisible items $O = \{o_1, \ldots, o_m\}$. Each agent $i$ has a complete and transitive strict preference order $\succ_i$ over $O$. The *ranking* of an item for an agent is defined as the sequence of the item in the agent's preference list. Specifically, an item's ranking in an agent's preference equals the number of items preferred by the agent plus one. An *allocation* is a partition of items into $n$ bundles $A = (A_1, \ldots, A_n)$, such that each agent $i$ is allocated a bundle $A_i$. We also denote $[p]$ as the set $\{1, \ldots, p\}$ for any positive integer $p$.

**Uncertainty in Preferences.** In this work, we allow agents to express uncertainty in their preferences in terms of "ties". Specifically, we adopt an uncertainty model considered in Aziz et al. [2016] and Aziz et al. [2019a] (termed as a

Table 1: Theorectical results

| | Weak SD Proportionality | SD Proportionality |
|---|---|---|
| FAIRPROB | in P (Thm 3.1) | in P (Thm 4.1) |
| EXISTSPOSSIBLYFAIR | in P (Thm 3.2) | in P (Thm 4.2) |
| EXISTSCERTAINLYFAIR | NP-Hard (Thm 3.3)
in P for constant $n$ (Thm 3.5)
in P for identical agents (Thm 3.4) | NP-Hard (Thm 4.3)
in P when the number of equivalent classes is at most 4 in all preferences. (Thm 4.4) |
| HIGHESTPROB | NP-hard (Thm 3.3)
in P for constant $n$ (Thm 3.5) | NP-hard (Thm 4.3) |

*compact indifference model* in their paper). In this model, each agent $i$ is allowed to report a weak preference list. For instance, consider such a preference list $(S_{i,1}) \succ_i \ldots \succ_i (S_{i,k_i})$. Each $S_{i,j}$ is a tied set in the weak preference list that we call the $j$th *equivalent class* of agent $i$. The actual strict preference $\succ_i$ of agent $i$ is then chosen uniformly at random from all linear order extensions of this weak order. The choices of the linear order extensions of different agents are independent. We denote $k_i$ as the number of equivalent classes of agent $i$.

**Example 1.** Consider the following allocation problem with 2 agents and 4 items: agent 1 has preference $(a, b) \succ_1 (c, d)$, agent 2 has preference $(a) \succ_2 (b, c, d)$. In this instance, the strict preference of agent 1 may be $a \succ_1 b \succ_1 c \succ_1 d$ or $a \succ_1 b \succ_1 d \succ_1 c$ or $b \succ_1 a \succ_1 c \succ_1 d$ or $b \succ_1 a \succ_1 d \succ_1 c$, each with probability $1/4$, while the strict preference of agent 2 may be $a \succ_2 b \succ_2 c \succ_2 d$ or $a \succ_2 b \succ_2 d \succ_2 c$, or the rest 4 preferences generated by the permutation of $b, c, d$, each with probability $1/6$.

**Proportional Fairness.** When only ordinal preferences are available, Aziz et al. [2015] defined several fairness notions that generalize proportionality to the ordinal setting. In this work we consider the following two definitions based on *stochastic dominance(SD)*.

- *Weak Stochastic Dominance (SD) proportionality*: an allocation satisfies *weak SD proportionality* if for each agent $i$, there exists $1 \le k \le m$ such that agent $i$ is allocated at least $\lfloor \frac{k}{n} + 1 \rfloor$ of her top $k$ items.

- *Stochastic Dominance (SD) proportionality*: an allocation satisfies *SD proportionality* if for each agent $i$ and $1 \le k \le m$, agent $i$ is allocated at least $\lceil \frac{k}{n} \rceil$ of her top $k$ items.

The above definitions of weak SD proportionality and SD proportionality are the simplifications of that in Aziz et al. [2015] when only considering strict preferences. This is when comparing a uniform allocation with another allocation of an agent using the original SD preference definition in Aziz et al. [2015], it reduces to the comparison between $\frac{k}{n}$ and the number of items allocated to this agent in her top $k$ favorites for every $1 \le k \le m$ under strict preferences.

**Example 1 (continued).** For our example above, when considering weak SD proportionality, one certainly fair allocation is to assign items $b, c, d$ to agent 1 and item $a$ to agent 2. However, when considering SD proportionality, this allocation has probability 0 to be fair for agent 2. The allocation that has the highest probability to be SD proportional is to assign $b, c$ to agent 1 and $a, d$ to agent 2, with fair probability $\frac{1}{4} \cdot \frac{2}{3} = \frac{1}{6}$.

## 3 WEAK SD PROPORTIONALITY

In this section, we consider the weak SD proportionality. First we show that given an allocation, one can efficiently compute the probability that this allocation is weak SD proportional.

**Theorem 3.1.** FAIRPROB *with regard to weak SD proportionality can be solved in polynomial time.*

*Proof.* Because the uncertainty in each agent's preference is independent, it suffices to show that the probability that the fairness condition is met for each agent $i \in N$ can be computed in polynomial time. In the following, we use dynamic programming to calculate this probability.

For one specific agent $i \in N$, there are $K := k_i$ equivalent classes and the $j$-th equivalent class has $s_j := |S_{i,j}|$ items. Firstly, we calculate the number of items owned by $i$ in the $j$-th equivalent class of $i$ and we denote it by $r_j \le s_j$.

The key to this dynamic programming algorithm is to identify the subproblem structure. We use $P_{\mathsf{num,tot,dis}}$ to represent the probability that there are tot items owned by agent $i$ arranged in the top num items with a fair verification parameter dis defined as follows:

$$\mathsf{dis} = \begin{cases} 0, & \text{if it's tranferred from the state with } \mathsf{dis}' = 0 \\ \max\left\{0, \left\lfloor \frac{\mathsf{num}}{n} + 1 \right\rfloor - \mathsf{tot}\right\}, & \text{otherwise} \end{cases}$$

Intuitively, dis shows the distance to weak SD proportionality and once it reaches 0, it will keep at 0.

The complete algorithm is shown in Algorithm 1. The key steps in this algorithm are the state transfers in Steps 3-21. We let sta represent the equivalent class that the $(\mathsf{num} + 1)$-ranked item belongs to, and let $p$ represent the probability

**Algorithm 1** FAIRPROB Algorithm

**Require:**
   $m$: number of items
   $n$: number of agents
   $K$: number of equivalent classes of agent $i$
   $s_{1,\ldots,K}$: number of items in each equivalent classes
   $r_{1,\ldots,K}$: number of items owned by agent $i$ in each equivalent classes
1: $P_{\text{num,tot,dis}} \leftarrow 0 \quad \forall 0 \leq \text{num}, \text{tot} \leq m, 0 \leq \text{dis} \leq \lfloor \frac{m}{n} + 1 \rfloor$
2: $P_{0,0,1} \leftarrow 1$
3: **for** num $\leftarrow 0$ to $m-1$ **do**
4:   **for** tot $\leftarrow 0$ to num **do**
5:     **for** dis $\leftarrow 0$ to $\lfloor \frac{m}{n} + 1 \rfloor$ **do**
6:       **if** $P_{\text{num,tot,dis}} \neq 0$ **then**
7:         Let sta be the minimum integer such that $\sum_{j \in [\text{sta}]} s_j \geq \text{num} + 1$.
8:         PrevP $\leftarrow P_{\text{num,tot,dis}}$
9:         $p \leftarrow \frac{\sum_{j \in [\text{sta}]} r_j - \text{tot}}{\sum_{j \in [\text{sta}]} s_j - \text{num}}$
10:         **if** dis $= 0$ **then**
11:           $P_{\text{num}+1,\text{tot}+1,\text{dis}}$ += PrevP $\cdot p$
12:           $P_{\text{num}+1,\text{tot},\text{dis}}$ += PrevP $\cdot (1-p)$
13:         **else**
14:           Let id be 1 if num $+ 1$ is a multiple of $n$ and
0 otherwise.
15:           $P_{\text{num}+1,\text{tot}+1,\max\{0,\text{dis}+\text{id}-1\}}$ += PrevP $\cdot p$
16:           $P_{\text{num}+1,\text{tot},\text{dis}+\text{id}}$ += PrevP $\cdot (1-p)$
17:         **end if**
18:       **end if**
19:     **end for**
20:   **end for**
21: **end for**
22: **return** $P_{m,\sum_{j \in [K]} r_j, 0}$

of the item with ranking (num $+ 1$) owned by $i$ under the present state in Steps 7-9.

We consider the specific transfers in two cases. Steps 11-12 solve the first case with dis $= 0$ where we have already reached fairness and the remaining is to arrange the order of the rest items. Steps 14-16 solve the second case with dis $> 0$ where dis has never met 0 before and we need to update it based on the definition. In both these two cases, we need to update the state based on the situation of whether the next item is owned by agent $i$. Specifically, when dis $= 0$, if the next item is owned by agent $i$, there are tot $+ 1$ items owned by $i$ arranged in the top num $+ 1$ items, so we transfer the state to $P_{\text{num}+1,\text{tot}+1,\text{dis}}$ in Step 11, otherwise only tot items in $i$'s bundle are arranged in the top num $+ 1$ items, which corresponds to the state $P_{\text{num}+1,\text{tot},\text{dis}}$ in Step 12. For the case of dis $> 0$, on top of the change of tot, we also use id to record the change of the term $\lfloor \frac{\text{num}}{n} + 1 \rfloor$ in the above

definition of dis in Step 14. Then, we transfer the state to $P_{\text{num}+1,\text{tot}+1,\max\{0,\text{dis}+\text{id}-1\}}$ in Step 15 if the next item is owned by $i$, or $P_{\text{num}+1,\text{tot},\text{dis}+\text{id}}$ in Step 16 otherwise.

Finally, Step 22 returns the probability that the fairness condition is met for agent $i$ with her all owned items. $\square$

We further show that EXISTSPOSSIBLYFAIR can also be solved in polynomial time based on Theorem 7 in Aziz et al. [2015]. We defer the detailed proof to the online supplement.

**Theorem 3.2.** EXISTSPOSSIBLYFAIR *with regard to weak SD proportionality can be solved in polynomial time.*

Next we turn to EXISTSCERTAINLYFAIR problem. Unlike EXISTSPOSSIBLYFAIR, we show that it is NP-hard to determine whether there exists an allocation that is weak SD proportional with probability one. To prove this hardness result, we reduce from a known NP-hard problem denote as (2,2)-E3-SAT [Berman et al., 2004].

---

**(2,2)-E3-SAT:** Given a boolean formula in conjunctive normal form in which each clause has three literals and each variable occurs exactly twice positive and twice negative, decide whether this boolean formula is satisfiable.

---

**Theorem 3.3.** EXISTSCERTAINLYFAIR *and* HIGHEST-PROB *with regard to weak SD proportionality are NP-hard.*

*Proof.* We reduce from (2,2)-E3-SAT to EXISTSCERTAIN-LYFAIR, which can also imply NP-hardness for HIGHEST-PROB. Considering a (2,2)-E3-SAT instance $F$ with $s$ variables $X = \{x_1, \ldots, x_s\}$ and $t$ clauses which satisfies $4s = 3t$. Based on $F$ we construct a problem instance of EXISTSCERTAINLYFAIR with $6s$ agents and $12s$ items.

The agents are divided into two sets. The first is a set of $2s$ agents $A = \{A_1, \ldots, A_{2s}\}$ where $A_{2k-1}$ corresponds to the true valuation of $x_k$ and $A_{2k}$ corresponds to the false valuation of $x_k$ for each $k \in [s]$. The second set is the set of the remaining $4s$ dummy agents $A' = \{A'_1, \ldots, A'_{4s}\}$.

The $12s$ items are divided into five sets as follows:

- A set of $t$ items $C = \{C_1, \ldots, C_t\}$, where $C_i$ corresponds to the $i$-th clause in (2,2)-E3-SAT;

- A set of $t/2$ items $T = \{T_1, \ldots, T_{t/2}\}$, given to the agents corresponding to the correct value of variables together with the items in $C$;

- A set of $s$ items $S = \{S_1, \ldots, S_s\}$, compensated for the agents representing the wrong value of variables;

- A set of $6s$ dummy items $B = \{B_1, \ldots, B_{6s}\}$, where each 2 items are bound to $3s$ agents in $A'$;

- A set of $3s$ dummy items $Q = \{Q_1, \ldots, Q_{3s}\}$, where each 3 items are bound to the remaining $s$ agents in $A'$.

Next, we construct the preference lists for agents. For each agent $A_i \in A$, we denote $R_i$ as a set of items such that: for each $k \in [s]$, $R_{2k-1}$ and $R_{2k}$ consist of the two items in $C$ corresponding to the clauses containing the positive $x_k$ and the negative $x_k$ respectively. The preference of each agent $A_i$ in $A$ is: $S_{\lceil i/2 \rceil} \succ (R_i, T, B) \succ$ (others). The preference of each dummy agent $A'_i$ in $A'$ is: $(B) \succ$ (others).

An example of this construction is shown below.

---

**Example 2.** Consider a simple (2,2)-E3-SAT problem instance with $s = 3$ variables and $t = 4$ clauses:

1. $x_1 \vee x_2 \vee \neg x_3$ 2. $x_1 \vee \neg x_2 \vee x_3$

3. $\neg x_1 \vee x_2 \vee x_3$ 4. $\neg x_1 \vee \neg x_2 \vee \neg x_3$

The corresponding instance of EXISTSCERTAINLYFAIR has 18 agents and 36 items. The preference lists are as follows:

- Agent 1: $S_1 \succ (C_1, C_2, T_1, T_2, B_1, \ldots, B_{18}) \succ (S_2, S_3, C_3, C_4, Q_1, \ldots, Q_9)$
- Agent 2: $S_1 \succ (C_3, C_4, T_1, T_2, B_1, \ldots, B_{18}) \succ (S_2, S_3, C_1, C_2, Q_1, \ldots, Q_9)$
- Agent 3: $S_2 \succ (C_1, C_3, T_1, T_2, B_1, \ldots, B_{18}) \succ (S_1, S_3, C_2, C_4, Q_1, \ldots, Q_9)$
- Agent 4: $S_2 \succ (C_2, C_4, T_1, T_2, B_1, \ldots, B_{18}) \succ (S_1, S_3, C_1, C_3, Q_1, \ldots, Q_9)$
- Agent 5: $S_3 \succ (C_2, C_3, T_1, T_2, B_1, \ldots, B_{18}) \succ (S_1, S_2, C_1, C_4, Q_1, \ldots, Q_9)$
- Agent 6: $S_3 \succ (C_1, C_4, T_1, T_2, B_1, \ldots, B_{18}) \succ (S_1, S_2, C_2, C_3, Q_1, \ldots, Q_9)$
- Agent 7-18: $(B_1, \ldots, B_{18}) \succ (S_1, S_2, S_3, C_1, \ldots, C_4, T_1, T_2, Q_1, \ldots, Q_9)$

---

If the (2,2)-E3-SAT instance has a satisfying assignment, we can construct an allocation as follows:

- If variable $x_i$ is TRUE, we assign $S_i$ to $A_{2i}$, otherwise we assign $S_i$ to $A_{2i-1}$;
- For the $j$-th clause, suppose the first true term in it is about $x_i$. Then, if this term is positive $x_i$, we assign $C_j$ to $A_{2i-1}$, otherwise we assign $C_j$ to $A_{2i}$.
- For agents in $A$ who do not receive any $S_i$, we assign items in $T$ to them so that these agents can each receive exactly two items. We know this can be satisfied because $2s = t + t/2$.
- Finally, we assign two items in $B$ to each of the first $3s$ agents in $A'$ and assign three items in $Q$ to each of the remaining $s$ agents in $A'$ arbitrarily.

For the agents in $A$ who get an item in $S$, their fairness conditions are met because they get their unique most preferred

item respectively. Each of the remaining agents in $A$ gets exactly two items that are not least preferred by them, so their fair conditions are also met. Finally, agents in $A'$ also meet the fair conditions because either they receive two items that they do not prefer the least, or they receive three items. Thus, this allocation must satisfy weak SD proportionality.

On the other hand, assume there is an allocation that is weak SD proportional with probability one, we can construct a satisfying assignment for the (2,2)-E3-SAT instance. First, the agents in $A$ who do not receive their most preferred item must get at least two items to meet the fair condition. Because there are at most $s$ agents that can receive items in $S$, at least $s$ agents in $A$ need at least two items. Next, for the agents in $A'$, they can meet the fairness condition if and only if they receive either two items in $B$ or three items. Because of the limit of the number of items in $B$, at most $3s$ agents in $B$ can meet the fair condition with only two items.

We can consider the most optimistic situation: $s$ agents in $A$ each get their most preferred item in $S$ and the remaining $s$ agents in $A$ get exactly two items in their respective second equivalent class but not in $B$; $3s$ agents in $A'$ each get exactly two items in $B$ while the remaining $s$ agents in $A'$ each get three items. The number of items needed in total is $12s$, which means that this situation is necessary.

Therefore, for any $k \in [s]$, exactly one agent between $A_{2k-1}$ and $A_{2k}$ gets the corresponding $S_k$ while the other one needs to get two items from $C \cup T$, whose size is $t/2 + t = 2s$. This means each item in $C$ must be chosen by the agents in $A$, and this will lead to a satisfying assignment for (2,2)-E3-SAT with such value scheme: for each pair $A_{2k-1}$ and $A_{2k}$ with $k \in [s]$, we set $x_k = $ FALSE if $A_{2k-1}$ gets $S_k$ otherwise we set $x_k = $ TRUE.

---

**Example 2 (continued).** The above mentioned (2,2)-E3-SAT instance is satisfiable by setting $x_1 = $ TRUE, $x_2 = $ TRUE, $x_3 = $ FALSE. We can get the corresponding allocation for the EXISTSCERTAINLYFAIR instance:

- The agent representing the wrong valuation of each variable takes the corresponding item in $S$: agent $2, 4$ and $5$ takes their respective $S_1, S_2, S_3$.
- Each item representing the clause is taken by the agent representing the first true term: agent 1 takes $C_1, C_2$, agent 3 takes $C_3$, and agent 6 takes $C_4$, and $T_1, T_2$ are given to agent 3 and 6 as a compensation.
- Each two of $B_{1,\ldots,18}$ are assigned to agent $7 - 15$, and each three of $Q_{1,\ldots,9}$ are assigned to agent $16 - 18$.

One can check that this is a certainly fair allocation in our EXISTSCERTAINLYFAIR instance.

---

$\square$

Next we present several positive results and short discussions on solving EXISTSCERTAINLYFAIR and HIGHESTPROB for special cases.

**Theorem 3.4.** EXISTSCERTAINLYFAIR *with regard to weak SD proportionality can be solved in polynomial time when all agents have identical preferences.*

When all agents have identical preferences, we can derive an optimal greedy algorithm that assigns items to agents such that each agent receives consecutive items in a preference order from the most to the least preferred items. This demonstrates the polynomial-time solvability of the EXISTSCERTAINLYFAIR. In this analysis, the assumption of identical preferences is critical since it allows us to sort all items before the allocation, which is a crucial step in the follow-up analysis.

**Theorem 3.5.** EXISTSCERTAINLYFAIR *and* HIGHESTPROB *with regard to weak SD proportionality can be solved in polynomial time when the number of agents is constant.*

When the number of agents $n$ is a constant, we can derive the following exact algorithm. First, we enumerate the number of items assigned to each agent from each equivalent class. Next, we use a perfect matching algorithm to find a corresponding allocation scheme and apply FAIRPROB to calculate the fair probability. The total number of enumerations is in the order of $O(n^{4n^2+n+1})$ which is a constant. However, this is no longer true if $n$ is a superconstant. Therefore, the constant assumption is an important assumption to get a polynomial-time solvable algorithm.

# 4 SD PROPORTIONALITY

In this section, we turn our focus to SD proportionality. This is a stronger fairness requirement than weak SD proportionality. Similar to the weak SD proportionality case, we show that with regard to SD proportionality, FAIRPROB and EXISTSPOSSIBLYFAIR can be solved in polynomial time, while EXISTSCERTAINLYFAIR and HIGHESTPROB are both NP-hard. The techniques we use to prove these results are similar in spirit to those for weak SD proportionality. However, due to the difference between these two fairness concepts, adjustments need to be made to the proofs to make them work in the SD proportionality setting. For example, to prove the NP-hardness for EXISTSCERTAINLYFAIR and HIGHESTPROB with regard to SD proportionality, we need to reduce from another NP-hard problem known as X3C from Johnson and Garey [1979].

Below we list all the results in this section. The details of all proofs, except the proof of Theorem 4.4, are deferred to the supplementary material due to page limits.

**Theorem 4.1.** FAIRPROB *with regard to SD proportionality can be solved in polynomial time.*

**Theorem 4.2.** EXISTSPOSSIBLYFAIR *with regard to SD proportionality can be solved in polynomial time.*

**Theorem 4.3.** EXISTSCERTAINLYFAIR *and* HIGHESTPROB *with regard to SD proportionality are NP-hard.*

In the proof of Theorem 4.3, we can see even if the number of equivalent classes of each agent is 6, EXISTSCERTAINLYFAIR is still NP-hard.

**Theorem 4.4.** EXISTSCERTAINLYFAIR *with regard to SD proportionality can be solved in polynomial time if the number of equivalent classes of each agent is at most* 4.

*Proof.* SD proportionality with probability one is a very demanding condition. We first list some properties that such an allocation must satisfy. First, a direct corollary from Aziz et al. [2015] shows that such an allocation may exist only when $m = qn$ for some integer $q$ and each agent gets exactly $q$ items.

Second, each agent must get all items in her first equivalent class to meet the fair condition with $k = 1$. Each agent also cannot get any items in her last equivalent class. This is because, suppose agent $i$ gets at least one item in her last equivalent class, we can make this item her least favorite item, so we need to use the other $q - 1$ items to satisfy all fair conditions with $k \leq qn - 1$, which is impossible when $n > 1$. These two conditions also imply that the number of equivalent classes for each agent must be greater than 1 and the number of items in the first equivalent class must be no more than $q$ for a certainly fair allocation to exist.

We will utilize the above characterizations to determine the existence of a certainly SD proportional allocation. More specifically, we will reduce the EXISTSCERTAINLYFAIR problem to a max-flow problem in a flow network $G$. The construction is as follows. Let the flow network $G = (V, E)$ with a capacity function $c : E \mapsto \mathbb{R}^+$. $V$ consists of a source vertex $s$, a target vertex $t$, a vertex set $A$ defined later and a vertex set $B$ of $m$ vertices representing the $m$ items. For each vertex $j$ in $B$, we create an edge $(j, t)$ with capacity $c = 1$. Next, for each agent $i$, we construct its vertices in $A$ depending on the number $k_i$ of its equivalent classes.

- $k_i = 2$: Because we cannot give agent $i$ any items in her last equivalent class and each agent must get all items in her first equivalent class, we must have $|S_{i,1}| = q$, otherwise there is no certainly fair allocation. Thus, we create a vertex $i$ in $A$ representing agent $i$, and construct edges $(i, j)$ with capacity $c = 1$ for each vertex $j \in S_{i,1}$ in $B$. We also construct an edge pointing from the source $s$ to $i$ with capacity $c = q$.

- $k_i = 3$: Similar to the case $k_i = 2$, agent $i$ must get all items in $S_{i,1}$ and the remaining $q - |S_{i,1}|$ items from $S_{i,2}$. We can use the algorithm for FAIRPROB to check whether such an allocation can meet agent $i$'s fairness

condition with probability one. If not, it means there does not exist a certainly fair allocation. Otherwise, we create two vertices $i_1, i_2$ in $A$ for agent $i$.

- The first vertex $i_1$ has an edge $(s, i_1)$ with $c = |S_{i,1}|$, and edges $(i_1, j)$ with $c = 1$ for each vertex $j \in S_{i,1}$ in $B$;
- The second vertex $i_2$ has an edge $(s, i_2)$ with $c = q - |S_{i,1}|$, and edges $(i_2, j)$ with $c = 1$ for each vertex $j \in S_{i,2}$ in $B$.

- $k_i = 4$: This is the most complicated case. We assume that the numbers of items owned by agent $i$ in the first three equivalence classes are $x, y, z$. It must satisfies $x = |S_{i,1}|$ and $x + y + z = q$. Once the value of $x$ is fixed, we observe that the probability that agent $i$ meets the fair condition increases with $y$. So we can find the minimum $y' \leq \min\{q - x, |S_{i,2}|\}$ such that $i$ can meet her fair condition with probability one. If there exists no such $y'$, there is no certainly fair allocation for this instance. Otherwise, we know for all eligible $y \geq y'$, agent $i$ can always meet the fair condition with probability 1. Following this idea, we create three vertices $i_1, i_2, i_3$ in $A$ for $i$:

  - the first vertex $i_1$ has an edge $(s, i_1)$ with $c = x$, and edges $(i_1, j)$ with $c = 1$ for each vertex $j \in S_{i,1}$ in $B$;
  - the second vertex $i_2$ has an edge $(s, i_2)$ with $c = y'$, and edges $(i_2, j)$ with $c = 1$ for each vertex $j \in S_{i,2}$ in $B$;
  - the third vertex $i_3$ has an edge $(s, i_3)$ with $c = q - x - y'$, and edges $(i_3, j)$ with $c = 1$ for each vertex $j \in S_{i,2} \cup S_{i,3}$ in $B$. The intuition for this step is to allow the allocation to give more items to agent $i$ in $S_{i,2}$.

With this construction, one can check that if there exists a maximum flow with value $m = qn$, the corresponding allocation must be SD proportional with probability one, because it meets all fair conditions for each agent according to our analysis. □

**Remark:** The algorithm can not be directly generalized to the case when some agent $i$ has more than 4 equivalent classes. To see it, note that if we continue to use this approach and assume that the number of items owned by agent $i$ in her top four equivalent classes are $x, y, z, w$, we still have $x = |S_{i,1}|$ and $x + y + z + w = q$. But it is no longer true that the fair probability is monotone increasing with both $y$ and $z$. This is because the fair probability may decrease if $z$ decreases even when $y$ increases.

# 5 EXPERIMENTS

In this section, we focus on HIGHESTPROB with respect to weak SD proportionality given the importance of finding al-

locations with a high fair probability. Since we have shown this problem is NP-hard, we will only present several heuristic algorithms for it. The performance and computational efficiency are evaluated on both synthetic and real datasets.

## 5.1 ALGORITHMS

We design and evaluate four algorithms for HIGHESTPROB. The details and parameters used in each algorithm can be found in the supplementary material.

1. **BASELINE (B)**: This is a baseline algorithm that assigns each item to a random agent to get a random allocation.

2. **LOCALSEARCH (LS)**: This is a local search algorithm that repeatedly move one item from one agent to another until the allocation reaches a local optima.

3. **MATCHING (M)**: This is a multi-round matching algorithm that matches each agent to an item in each round. In particular, the weight of each arc linking an agent to an item is based on a function of the item's possible rankings in the agent's preference list. We apply multi-round maximum weight matching algorithms to assign the items to each agent.

4. **GREEDY (G)**: This algorithm first converts HIGHESTPROB to a submodular welfare maximization problem and then uses a greedy algorithm from Lehmann et al. [2006] to output a 2-approximation solution.

## 5.2 DATASETS

We test the above four algorithms on both synthetic and real datasets.

1. **Synthetic dataset**: We enumerate $n$ from 2 to 20 and $m$ from $n$ to $5n$. For each $p \in \{0.02, 0.03, 0.04, 0.05, 0.06\}$, we create 30 datasets in the following way: For each agent we create a random permutation of all items. For each pair of adjacent items in the permutation, we separate them into two equivalent classes with probability $p$. We choose $p$ to be in this range because when $p$ becomes larger, all the four algorithms can easily find an almost certainly fair assignment and there is no significant difference in their performance.

2. **Real dataset**: We use the Preflib database from Mattei and Walsh [2013], which is an online database of real-world preference profiles to test our algorithms. We select 11 datasets from two categories in this database:

   (a) *Matching Data (MD-00002)*: This category contains bidding preferences of reviewers over a subset of papers at Computer Science conferences. Each preference is an incomplete list with ties.

We convert each preference into a complete list with ties by adding the remaining items as the last equivalent class. This category has 3 datasets corresponding to 3 different conferences.

(b) *Matching Data (MD-00003)*: This category contains bids of students over a set of projects for student/project allocations at a university. It has 8 datasets in total, all with complete preferences with ties, with 31-51 students and 56-155 projects.

## 5.3  RESULTS AND DISCUSSIONS

### 5.3.1  Synthetic dataset

We measure the performance of each algorithm by the average fair probability across various tested datasets. Figure 1 shows the average fair probability generated by each algorithm for each $n$ from 2 to 20. Each average fair probability is taken over multiple datasets with different $m$ and $p$. Figure 2 presents the average running time of each algorithm.

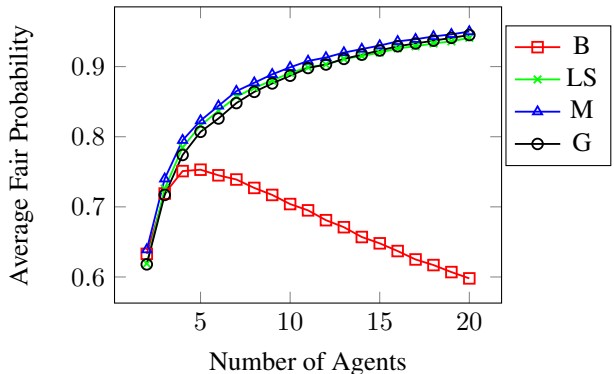

Figure 1: Algorithm performance over synthetic data

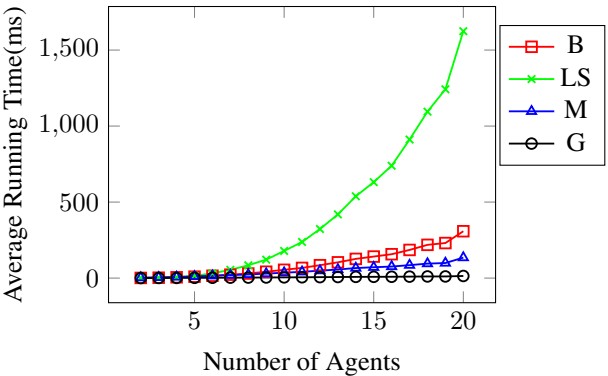

Figure 2: Algorithm running time over synthetic data

From the figures, one can see that LOCALSEARCH, MATCHING and GREEDY generate allocations with a much higher average fair probability than BASELINE when the number

of agents is at least 5. Besides, when the number of agents $n$ increases, the running time of LOCALSEARCH increases significantly, while BASELINE, MATCHING, and GREEDY keep low running time. In summary, GREEDY and MATCHING require the least running time and perform the best consistently for different values of $n$.

### 5.3.2  Real dataset

Table 2 presents the fair probabilities of the suggested allocations by each algorithm on 11 real datasets. Table 3 further provides their running times. In the real datasets, BASELINE still performs the worst. The other three algorithms can generate allocations that are fair with probability 1 for most cases, with two exceptions for LOCALSEARCH and one exception for GREEDY. MATCHING is the only algorithm that performs consistently well. In terms of the running time, LOCALSEARCH takes much longer than the other three. The BASELINE and MATCHING take a little more time than GREEDY. But it is still acceptable.

Table 2: Fair probability of all algorithms over real data (sorted by the number of agents $n$)

| no. | n | m | B | LS | M | G |
|---|---|---|---|---|---|---|
| 1 | 24 | 52 | 0.03 | 1.00 | 1.00 | 1.00 |
| 2 | 31 | 54 | 0.11 | 1.00 | 1.00 | 1.00 |
| 3 | 31 | 103 | 0.02 | 1.00 | 1.00 | 1.00 |
| 4 | 32 | 102 | 0.06 | 1.00 | 1.00 | 1.00 |
| 5 | 34 | 63 | 0.23 | 1.00 | 1.00 | 1.00 |
| 6 | 35 | 61 | 0.01 | 1.00 | 1.00 | 1.00 |
| 7 | 37 | 56 | 0.00 | 0.01 | 1.00 | 1.00 |
| 8 | 38 | 133 | 0.03 | 1.00 | 1.00 | 1.00 |
| 9 | 51 | 147 | 0.28 | 1.00 | 1.00 | 1.00 |
| 10 | 51 | 155 | 0.00 | 1.00 | 1.00 | 1.00 |
| 11 | 146 | 176 | 0.00 | 0.00 | 1.00 | 0.37 |

Table 3: Running time(s) of all algorithms over real data (sorted by the number of agents $n$)

| no. | n | m | B | LS | M | G |
|---|---|---|---|---|---|---|
| 1 | 24 | 52 | 0.074 | 0.908 | 0.058 | 0.003 |
| 2 | 31 | 54 | 0.078 | 1.017 | 0.075 | 0.005 |
| 3 | 31 | 103 | 1.008 | 29.54 | 0.212 | 0.024 |
| 4 | 32 | 102 | 1.127 | 40.02 | 0.264 | 0.025 |
| 5 | 34 | 63 | 0.143 | 0.738 | 0.120 | 0.006 |
| 6 | 35 | 61 | 0.172 | 1.798 | 0.140 | 0.006 |
| 7 | 37 | 56 | 0.126 | 2.531 | 0.131 | 0.006 |
| 8 | 38 | 133 | 2.148 | 63.73 | 0.388 | 0.039 |
| 9 | 51 | 147 | 2.410 | 9.612 | 0.618 | 0.039 |
| 10 | 51 | 155 | 3.250 | 480.7 | 0.687 | 0.055 |
| 11 | 146 | 176 | 3.559 | 9255 | 5.533 | 0.064 |

In conclusion, according to the results from both synthetic and real data, LOCALSEARCH has a good performance at a high computation cost. GREEDY has decent performance with the best computation efficiency. MATCHING is a balanced algorithm with consistently good performance and high computation efficiency.

# 6 CONCLUSIONS

In this paper, we present algorithmic and complexity results in computing fair allocations assuming agents' preferences over items are ordinal and have uncertainties. Under the proposed fair notions, we provide polynomial-time solvable algorithms to find the probability that a given allocation is fair and determine whether there exists an allocation with non-zero fair probability. We show that it is NP-hard to see whether there exists an allocation that is fair with probability one. Finally, we show that to find an allocation with the highest fair probability is NP-hard. We further provide several heuristics for this problem. The performance of the heuristics is examined thoroughly on both synthetic and real datasets. One possible direction for future work is to consider other fairness concepts in the context of ordinal and uncertain preferences and study approximation algorithms to find fair allocations. It is also interesting to combine the fairness notions with other properties such as stability in the uncertain ordinal preference setting.

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
