# OpenReview forum: "Proportional Allocation of Indivisible Resources under Ordinal and Uncertain Preferences"
_auai.org/UAI/2022/Conference — UAI 2022 Poster_

### Official Review · Reviewer_NgaG · 2022-04-06

**Q2(1) Originality/Novelty:** 2
**Q2(2) Significance/Impact:** 3
**Q2(3) Correctness/Technical Quality:** 3
**Q2(6) Clarity Of Writing:** 3
**Q6 Overall Score:** 6
**Q8 Confidence In Your Score:** 3

**Q1 Summary And Contributions:**

The authors consider the problem of resource allocation between agents when their preferences are ordinal and uncertain. They focus on the notion of (weak) stochastic dominance proportionality and analyze the computational complexity of four problems, as well as the performance of a number of algorithms.

**Q2 Assessment Of The Paper:**

More detailed information regarding each of these aspects is given below:

**Q2(4) Quality Of Experiments (Optional):**

3: Good: The experimental evaluation is adequate, and the results convincingly support the main claims.

**Q2(5) Reproducibility:**

3: Good: Key resources (e.g., proofs, code, data) are available and key details (e.g., proofs, experimental setup) are sufficiently well-described for competent researchers to confidently reproduce the main results.

**Q3 Main Strengths:**

The authors show a good knowledge of the topic at hand, and the results seem to be new and interesting. The experimental part is detailed and the paper is well-written.

**Q4 Main Weakness:**

The main assumption of the paper is that the preferences are ordinal and complete, which leads to imposing a uniform distribution over the set of items an agent is indifferent on. This assumption of a complete order is arguably too strong, as there will be cases where an agent cannot express a preference between two items without being indifferent between them

In addition, the paper focuses on (weak) stochastic dominance proportionality, but other options are not discussed and the possible relevance of the other possibilities considered in Aziz (2015) is not discussed.

**Q5 Detailed Comments To The Authors:**

While I believe the results in the paper to be both correct and relevant, I also think that the motivation of the paper and the choices made by the authors could be improved. The literature on this topic is very large and, while the problems considered are interesting, there are other notions of fairness that could be considered. A similar comment could be made on the problems that have been considered.

The other main comment is about the assumption of a complete order, that is perhaps too strong in some contexts. If we allowed for some sets where an incomparability between the items is possible, then we could consider other probability distributions instead of a uniform one (possibly a set of distributions).

On the other hand, the cases where the complexity of the EXISTSCERTAINLYFAIR  and the HIGHESTPROB algorithms is polynomial seem to be too specific to be useful in applications. I would suggest also to include a discussion of the extent to which the results can(not) be improved, similarly to what has been done in the remark at the end of Section 4.

**Q7 Justification For Your Score:**

The paper is fine but the choices made on the proportionality criteria and the algorithms are a bit too specific and could be justified better.

**Q9 Complying With Reviewing Instructions:**

1: Yes.

---

### Official Review · Reviewer_wyPc · 2022-04-10

**Q2(1) Originality/Novelty:** 3
**Q2(2) Significance/Impact:** 2
**Q2(3) Correctness/Technical Quality:** 3
**Q2(6) Clarity Of Writing:** 4
**Q6 Overall Score:** 7
**Q8 Confidence In Your Score:** 3

**Q1 Summary And Contributions:**

The paper deals with the allocation of indivisible items to agents, when the preferences of agents over items are strict weak ordering. Incompleteness can be viewed as a form of uncertainty, where each linear extension of the strict weak ordering is considered as possible and taken into account in the determination of a fair allocation. Complexity results are established for various computational problems related to fair allocation in this setting, and numerical experiments are provided.

**Q10 Ethical Concerns (Optional):**

No ethical concern.

**Q2 Assessment Of The Paper:**

More detailed information regarding each of these aspects is given below:

**Q2(4) Quality Of Experiments (Optional):**

3: Good: The experimental evaluation is adequate, and the results convincingly support the main claims.

**Q2(5) Reproducibility:**

3: Good: Key resources (e.g., proofs, code, data) are available and key details (e.g., proofs, experimental setup) are sufficiently well-described for competent researchers to confidently reproduce the main results.

**Q3 Main Strengths:**

* The paper is well-written and the tackled problems are clearly defined.
* The technical results are non-trivial and well explained.
* Numerical experiments are provided to evaluate heuristics for the problems proved NP-hard.

**Q4 Main Weakness:**

* The contributions are a bit incremental compared to the case where there are ties between the items.
* The baseline algorithm is not a strong contender; it would have been interesting to provide an (exponential) exact procedure for the determination of an allocation that is fair with the highest probability, even if the problem is NP-hard, and to evaluate the quality of the allocations returned by the heuristics.

**Q5 Detailed Comments To The Authors:**

The fairness criteria considered in the paper are weak Stochastic Dominance (SD) proportionality and SD proportionality.

Four problems are studied in the paper, namely 1) the evaluation of the probability that an allocation is fair, 2) the problem consisting in deciding whether there exists a fair allocation (w.r.t. the criterion of weak SD proportionality or SD proportionality) with non-zero probability, 3) the problem consisting in deciding whether there exists a fair allocation (w.r.t. the criterion of weak SD proportionality or SD proportionality) with probability one, 4) the problem of determining an allocation that is fair with the highest probability.

Note that the addition of examples illustrating the notions of weak SD proportionality and SD proportionality without uncertainty would be welcome.

For the criterion of weak SD proportionality, it is shown that problem 1 can be solved in polynomial time by a dynamic programming procedure. Problem 2 is also solvable in polynomial time, while problems 3 and 4 are NP-hard (by reduction from (2,2)-E3-SAT).

For the criterion of SD proportionality, it is shown that problems 1 and 2 can also be solved in polynomial time, while problems 3 and 4 are NP-hard (by reduction from X3C). Nevertheless, problem 3 can be solved in polynomial time if the number of equivalence classes is at most 4 for problem 3, by formulating it as a max-flow problem in a network. The numerical experiments are conducted on synthetic data, as well as real-world data from PrefLib.

Finally, numerical experiments are presented to compare various heuristics for problem 4 w.r.t. weak SD proportionality.

Typos
P.1: a similar way in -> a similar way as in.
P.6: from Johnson and Gary -> from Garey and Johnson.
P.6 above Theorem 4.4: even the number -> even if the number.
P.7: Algo 3 and 4 would be hard to grasp precisely without the supplementary material; in my opinion, a link to online supplementary material should be provided in a final version of the paper.

**Q7 Justification For Your Score:**

The strong technical contribution outweighs the small weakness in the numerical experiments (in the evaluation of the heuristics). Numerical experiments are moreover welcome.

**Q9 Complying With Reviewing Instructions:**

1: Yes.

---

### Official Review · Reviewer_gNC9 · 2022-04-12

**Q2(1) Originality/Novelty:** 3
**Q2(2) Significance/Impact:** 3
**Q2(3) Correctness/Technical Quality:** 3
**Q2(6) Clarity Of Writing:** 3
**Q6 Overall Score:** 6
**Q8 Confidence In Your Score:** 3

**Q1 Summary And Contributions:**

This paper studies a fair resource allocation problem assuming agents’ preferences over items are ordinal and have uncertainties. To this end, fair notions and polynomial-time solvable algorithms are proposed with experimental results on both synthetic and real datasets.

**Q2 Assessment Of The Paper:**

More detailed information regarding each of these aspects is given below:

**Q2(4) Quality Of Experiments (Optional):**

3: Good: The experimental evaluation is adequate, and the results convincingly support the main claims.

**Q2(5) Reproducibility:**

3: Good: Key resources (e.g., proofs, code, data) are available and key details (e.g., proofs, experimental setup) are sufficiently well-described for competent researchers to confidently reproduce the main results.

**Q3 Main Strengths:**

1. The assumption of agents’ preferences over items are ordinal and have uncertainties which is practical.
2. Corresponding fair notions and algorithms.
3. The paper is well-written and is easy to follow.

**Q4 Main Weakness:**

Although the authors discuss the difference from the previous works, e.g., Aziz et al. [2019a] and Aziz et al. [2016], I feel this work is somewhat incremental, ranging from uncertain preferences, Pareto optimal, etc.

**Q5 Detailed Comments To The Authors:**

Can we further relax the uncertain preferences constraint as in Fair Decision-making Under Uncertainty at ICDM'21?

**Q7 Justification For Your Score:**

Check above.

**Q9 Complying With Reviewing Instructions:**

1: Yes.

---

### Decision · Program_Chairs · 2022-05-15

**Decision:**

Accept (Poster)

**Comment:**

Meta Review: The reviewers all concur that this is a nicely done paper, which provides non-trivial results about an interesting setting.

The main criticism made by the reviewers is that this work is, to some extent, of incremental nature. Authors should therefore make an effort to emphasize the significance of their results.